

# Soil temperatures and active carbon components as key drivers of C stock dynamics between two different stand ages of *Larix principis-rupprechtii* plantation

Junyong Ma, Hairong Han and Xiaoqin Cheng

Key Laboratory of Ministry of Forest Cultivation and Conservation of Ministry of Education, Beijing Forestry University, Beijing, China

## ABSTRACT

Forest soils sequester a large amount of carbon (C) and have a significant effect on the global C balance. Forests are commonly managed to maintain certain age structures but the effects of this management on soil C pools (kg C m$^{-2}$) is still uncertain. We compared 40-year-old (1GF) and 24-year-old (2GF) plantations of *Larix principis-rupprechtii* in North China. Specifically, we measured environmental factors (e.g., soil temperature, moisture, and pH), the active C and nitrogen (N) pools (e.g., soil organic C, soil total N, dissolved organic C and N, microbial biomass C and N), and soil processes (e.g., C mineralization and microbial activity in different seasons) in five soil layers (0–50 cm, 10 cm for each soil layer) across the growing seasons in three 25 m × 25 m plots in each age class (1GF and 2GF). Findings indicated that the soil organic C pool in the older 1GF forest (12.43 kg C m$^{-2}$) was significantly higher than 2GF forests (9.56 kg C m$^{-2}$), and that soil temperature in 1GF forests was 9.8 °C, on average, 2.9% warmer than temperature in 2GF forests. The C lost as carbon dioxide ($CO_2$) as a result of mineralization in the 2GF plots may partly explain the lower soil organic C pool in these younger forests; microorganisms likely drive this process.

## INTRODUCTION

Globally soils hold 411~2111 Pg C (*Tian et al., 2015*). This storage and the stabilization of soil carbon (C) pools is crucial for the atmospheric $CO_2$ balance (*Paustian et al., 2016*) and for soil structure, biological activity, and nutrient and water cycles, all of which lead to more productive soils (*Lal, 2011*; *Seremesic et al., 2011*). Plants play a critical role in the C cycle; moving C from external matter and energy to soil, mostly by fixing C through photosynthesis and releasing C through litter or root decomposition back to the atmosphere as $CO_2$ (*Yang et al., 2007*; *Kim et al., 2011*). In terrestrial ecosystems, soil C pools account for about half of the C, with forest soils accounting for much of that C storage (*Post et al., 1982*). Soil C pools act as both C sources and sinks in their regulation of the global C balance.

Corresponding author
Hairong Han,
hanhr6015@bjfu.edu.cn

Understanding the effects of management practices, including afforestation, on C and nutrient cycling in forest ecosystems is critical to anticipating ecosystem changes and improving climate change resilience (*Magruder, Chhin & Palik, 2013*; *Johnson, Chhin & Zhang, 2017*). Many forests, especially plantation forests, are managed for particular age structures. Forest age influences many aspects of forest structure and function, including vertical structure (*Bao, Qiuliang & Liming, 2015*), net primary productivity, and net ecosystem productivity (*Pregitzer & Euskirchen, 2010*; *Hember et al., 2012*; *Lunstrum & Chen, 2014*; *Wang, Yao & Wang, 2018*). Recent research estimated that older natural forests have better soil quality (*Lucas-Borja et al., 2016*). With the varying of forest ages, the composition, biomass accumulation and distribution of nitrogen in forests changes and may affect soil microorganisms indirectly by changing soil temperature, water content or litter (*Litton et al., 2003*; *Jia et al., 2005*). Forest soil carbon cycles are also influenced by microorganisms (*Throckmorton et al., 2008*; *Leff, Nemergut & Grandy, 2012*), forest soil microbial abundance and biomass and diversity (*Tian et al., 2015*).

The magnitude of a forest C pool is determined by a combination of environmental factors and microorganisms, commonly represented by the empirical relationship between C sink and stand age and environmental factors (*Law et al., 2002*; *Hember et al., 2012*). Many studies have explored the impacts of human activities on environmental factors and, in turn, forest function and dynamics (*Naether et al., 2012*; *Mayor et al., 2017*). These studies have identified many environmental variables with close relationships to soil organic C (SOC) dynamics in forests systems, including soil moisture (*Singh et al., 2014*), pH (*Hobara, Kushida & Kim, 2016*), roots and associated fungi (*Clemmensen, Bahr & Ovaskainen, 2013*), above-ground vegetation biomass, species diversity (*Will et al., 2015*) and litter production (*D'Orazio, Traversa & Senesi, 2014*), disturbance by management (*Johnson & Curtis, 2001*), and ambient temperature (*Lutz, Shugart & White, 2013*). The complexity of how forest soil C pools vary as a result of changes in environmental factors creates many uncertainties (*Hoffmann, Hoffmann & Johnson, 2014*; *Hugelius et al., 2014*; *Sierra, Malghani & Loescher, 2017*). We believe that researchers should account for environmental variables when studying C process under different stand ages.

Except for being affected by environmental variables, forest C pools can also be affected by the dynamics of active C components (*Knorr et al., 2005*). Of the total mineral SOC, 5% belongs to the labile fraction; a 1% reduction in soil labile organic C (LOC) pool could result in an efflux of $0.75 \times 10^{15}$ g of C to atmosphere (*Zou et al., 2005*). LOC is more sensitive to short-term land use change than SOC (*Yang et al., 2009*; *Liang et al., 2012*), thus LOC is considered an early indicator of changes in C dynamics in the forest ecosystem (*Franzluebbers & Arshad, 1997*). Understanding how LOC responds to different forest management regimes is crucial for revealing C dynamics and the mechanisms influencing long-term C storage.

Soil processes (e.g., dissolved organic carbon sensitivity to season, SOC mineralization, microbial activity) may also respond differently depending on forest stand age. Operationally, SOC is analyzed according to chemical fractions to indicate soil processes. Dissolved organic C (DOC) is vital to soil nutrient cycling, especially the SOC stock; DOC is active in the process of physical movement and chemical transformation of C and

nutrients in soil (*Chen & Xu, 2008*). Studies have indicated that DOC is also sensitive to seasonal changes, soil properties, fertilizer addition, and land use change, and that DOC is intermediately linked to SOC mineralization (*Cheng et al., 2001*), and therefore is considered one of the most important pathways of soil C loss in forest ecosystems (*Gielen, Neirynck & Luyssaert, 2011*).

Microbial biomass C (MBC), is a sensitive biological C component and has been widely studied in forest ecosystems to understand microbial activity, C stock (*Xu, Inubushi & Sakamoto, 2006*; *Michelsen, Andersson & Jensen, 2004*), and environmental change (*Williams, Rice & Owensby, 2000*). Turnover of SOC is mediated by soil organisms; a number of studies have linked the catabolic diversity of soil microorganisms to SOC content as well as DOC (*Marschner & Kalbitz, 2003*; *Cookson et al., 2005*).

Mineralization of C ($CO_2$ production) is also essential in modeling soil C dynamics and ecosystem responses to changing environmental factors (*Schlesinger & Andrews, 2000*; *Stewart et al., 2008*). Soil incubation is widely used as a direct approach to quantify mineralizable soil C (*Ahn et al., 2009*). The abundance, structure and activity of soil microbial communities play important roles in SOC mineralization and sequestration (*Six, Frey & Thiet, 2006*). Studies of soil microbe environments—specifically, microbial community structure, determined by phospholipid fatty acid analysis (PLFA)—have found that tree plantations established on cropland or through afforestation could influence soil microbial communities and ecosystem sustainability (*Ding et al., 1992*; *Lauber et al., 2008*).

To provide a reference of C dynamics for forests of different ages, we established permanent plots in older (40 years) and younger (24 years) *Larix principis-rupprechtii* plantations in the montane secondary forest of Shanxi Province, North China. In this region, large areas of *Larix principis-rupprechtii* plantations have been managed by the local forestry administration since the 1970s. In these plots we measured a number of important variables related to C dynamics, with the aim to: (i) measure C stocks in forests of different ages; (ii) monitor important environmental factors; (iii) identify factors that might drive differences in C dynamics in the older and younger forests; and (iv) evaluate strategies for storing C in managed forests. We hypothesized that older forests soils have more C stored in soils, that the younger forests release more C derived from its active C components—such as DOC, MBC and mineralized C (MC)—and that the active C components interact with each other.

## MATERIALS AND METHODS

### Experimental site

The study site was at Taiyue Mountain, Shanxi Province, North China (112°00′47″E, 36°47′05″N; 112°01′~112°15′E, 36°31′~36°43′N; elevation 2000–2359 m above sea level). The research was conducted at the Taiyue Mountain Ecosystem Research Station for fieldwork. We studied *Larix principis-rupprechtii* plantations of two ages: the "first-generation forest" or 1GF (40 years old), which was planted and remained unchanged since 1976; and the "second generation forest" or 2GF (24 years old), which was first planted since 1976, then clear cut in 1991, and planted as new *Larix principis-rupprechtii* forest in 1992.

**Table 1 Basic characteristics of the plots.** Soil characteristics of the studied stands for the average of 0–50 cm, standard error is presented. DBH: diameter at breast height; BD: bulk density. All the basic information was measured in August of 2016 (means ± SD, $n = 3$).

| Treatment | Plot | Age (years) | Altitude (m) | Slope (°) | Aspect | Tree height (m) | DBH (cm) | Living branch height (m) | BD (g cm$^{-3}$) | Mechanical composition (%) | | |
|---|---|---|---|---|---|---|---|---|---|---|---|---|
| | | | | | | | | | | <0.002 mm | 0.002~0.05 mm | 0.05~2.00 mm |
| 1GF | Plot. 1 | 40 | 2,050 | 22.5° | North | 14.55 ± 3.7 | 18.85 ± 7.00 | 4.98 ± 2.90 | 1.18 | 25.83 | 35.40 | 38.77 |
| | Plot. 2 | 40 | 2,040 | 17° | North | 14.81 ± 4.10 | 20.86 ± 8.00 | 4.96 ± 2.70 | 1.30 | 18.30 | 35.00 | 46.70 |
| | Plot. 3 | 40 | 2,060 | 18° | North | 16.25 ± 2.50 | 22.22 ± 7.00 | 4.36 ± 2.00 | 1.32 | 22.07 | 29.90 | 48.03 |
| 2GF | Plot. 4 | 24 | 2,010 | 17° | North | 15.8 ± 3.53 | 12.83 ± 3.12 | 5.2 ± 1.24 | 1.39 | 17.09 | 30.00 | 52.91 |
| | Plot. 5 | 24 | 2,022 | 16° | North | 12.56 ± 2.00 | 13.07 ± 3.20 | 6.08 ± 1.40 | 1.37 | 18.27 | 31.93 | 49.80 |
| | Plot. 6 | 24 | 2,043 | 19° | North | 12.18 ± 1.80 | 13.64 ± 4.40 | 6.85 ± 1.00 | 1.22 | 16.11 | 37.32 | 46.57 |

**Table 2 Main plant species diversity in shrub and herb layer of the plots and litter thickness across sampling seasons in 2017.** The plant species diversity was investigated in August of 2016.

| | | Shrub layer | | | Herb layer | | | Litter thickness in different sampling seasons in 2017 | | | | |
|---|---|---|---|---|---|---|---|---|---|---|---|---|
| | | Species richness | Species Shannon index | Species evenness | Species richness | Species Shannon index | Species evenness | Apr | Jun | Aug | Oct | Average |
| 1G | Plot. 1 | 8 | 1.77 | 0.85 | 39 | 3.42 | 0.93 | 5.02 | 3.21 | 4.52 | 7.21 | 4.99 |
| | Plot. 2 | 4 | 1.32 | 0.95 | 28 | 3.15 | 0.94 | 6.52 | 3.92 | 3.50 | 6.42 | 5.09 |
| | Plot. 3 | 4 | 1.24 | 0.89 | 27 | 3.09 | 0.94 | 5.53 | 3.76 | 4.93 | 7.12 | 5.34 |
| 2G | Plot. 4 | 4 | 1.33 | 0.96 | 24 | 2.96 | 0.93 | 4.38 | 2.01 | 4.32 | 5.21 | 3.98 |
| | Plot. 5 | 6 | 1.79 | 1.00 | 25 | 3.09 | 0.96 | 4.53 | 2.92 | 2.56 | 5.33 | 3.84 |
| | Plot. 6 | 7 | 1.83 | 0.94 | 25 | 3.04 | 0.95 | 3.91 | 3.10 | 3.33 | 5.80 | 4.04 |

In August 2016, we established three 25 m × 25 m plots in each of the 1GF and 2GF forests. We left 5-m gaps between plots. The plots were sited in locations that were similar in elevation, slope, and aspect (Table 1). The composition of understory plants was also similar across sites (Table 2). At each plot, a UTBI-001 (HOBO) was embedded in the soil at 10-cm depth (plant litter was removed before the UTBI-001 embedded then backfilled), and temperature was recorded every 30 min between April 20 and October 20, 2017. The study plot soil type is Haplic luvisols (*FAO, 2015*) with a thickness of 50–110 cm.

### Soil sampling and background survey

In August 2016, we collected dug three soil profiles in each plot and collected soil samples from five depths (0–50-cm, one sample each at each 10-cm depth). We used a ring knife method to measure soil bulk density for each section, consistent with *Dam et al. (2005)*. In April, June, and August 2017, we collected soil samples with an auger, same depth with August 2016. We collected nine soil cores from each plot and mixed them to form one composite sample per plot. We measured litter thickness when we collected the soil samples. The samples were stored at 4 °C in plastic bags for a few days before analysis. To homogenize the soil material, we sieved the samples through a 2-mm sieve, which removed live roots, mycorrhizal mycelia and coarse plant remnants. We then

divided the soil samples into three subsamples for chemical analysis, determination of DOC and MBC, and cumulative C mineralization. The subsamples for chemical analysis were air dried and further subdivided: subsamples for pH analysis were filtered through a 2-mm sieve, while subsamples for other chemical analyses (e.g., C and N concentrations) were filtered through a 0.15 mm sieve and stored at room temperature. The subsamples for determination of DOC and MBC were stored at 4 °C and analyzed within 2 weeks. The subsamples for cumulative C mineralization were stored at 4 °C, analyzed within 72 h, and incubated at 25 °C for 1 week.

## Laboratory analysis

C and N concentrations were measured directly from a subset of air-dried samples by an elemental analyzer (Thermo Scientific FLASH 2000 CHNS/O; Waltham, MA, USA).

MBC concentration was measured using an $HCl_4$–fumigation extraction technique: $10 \pm 0.5$ g of fresh soil was fumigated with $HCl_4$, then extracted with 40 mL of 0.5 mol·$L^{-1}$ $K_2SO_4$, shaken for 1 h at 350 r $min^{-1}$, and filtered through a 0.45 μm membrane after centrifuging 5 min at 3000 r $min^{-1}$. We quantified the filtrate concentration using a total organic C analyzer (Multi N/C 3000, Germany). We measured the DOC concentration using the carbon concentration of 0.5 mol·$L^{-1}$ $K_2SO_4$ extract method: extracted with 40 mL of 0.5 mol·$L^{-1}$ $K_2SO_4$, shaken for 1 h at 350 r $min^{-1}$, and filtered through a 0.45 μm membrane after centrifuging 5 min at 3000 r $min^{-1}$. We quantified the filtrate concentration using a total organic C analyzer (Multi N/C 3000, Germany) (*Boyer & Groffman, 1996*).

MBC was calculated as:

$$MBC = EC/k_{EC1} \tag{1}$$

$$MBC = EC/k_{EC2} \tag{2}$$

where EC = (organic C, N extracted from fumigated soils) − (organic C, N extracted from non-fumigated soils), and $k_{EC1} = 0.38$, $k_{EC2} = 0.54$

Total pools of N and C as well as active C and N components were calculated using the formula:

$$\text{Pool} \left(kg \ m^{-2}\right) = \text{Concentration} \left(g \ kg^{-1}\right) * BD_{<2} \left(g \ cm^{-3}\right) * \text{layer thickness(cm)} \tag{3}$$

We measured soil C mineralization using an incubation method. We pre-incubated field samples (50 g fresh soil samples) for 1 week in 300-ml sealed jars at 25 °C in darkness to reduce the effects of necrosis generated by sampling and sample manipulation. We then further incubated samples at 25 °C in darkness; we considered the first day after pre-incubation to be day 1 of the incubation period. The $CO_2$–C evolved from soils over the incubation period (7, 14, 21, 28, 35, 42, 49 and 56 days) was trapped in 0.1 mol $L^{-1}$ NaOH, and the remaining NaOH was determined by titrating using 0.05 mol $L^{-1}$ HCl after precipitation of carbonate with 1 ml 1 mol $L^{-1}$ $BaCl_2$. The cumulative $CO_2$–C (mg C $kg^{-1}$) was calculated by the cumulative production of $CO_2$ from the soils during the 56-day

pre-incubation plus incubation and expressed as milligrams of $CO_2$–C per kilogram dry soil. When the value is normalized to SOC content, the quantity of mineralized C (% of SOC, mg C kg $SOC^{-1}$) is given by the equation:

$$Mineralized\ C\ =\ Cumulative\ C\ mineralization/SOC \tag{4}$$

where C is the SOC content (g $kg^{-1}$).

We analyzed soil texture (mechanical composition) using the pipette method (*Gee & Bauder, 1986*). Air-dried soil samples that had been passed through a 2-mm sieve were mixed 1:2.5 (soil : water) and tested with a pH meter (Sartorius PB-10). Gravimetric soil water concentration was measured as mass loss of fresh soil (10 ± 0.5 g) after drying for 24 h at 105 °C in aluminum cans.

We measured microbial community structure using PLFA. We extracted PLFAs using a modified procedure and analyzed them using the methods of *Moore-Kucera & Dick (2008)*. We calculated PLFAs based on a 19:0 internal standard content. We classified the following soil microbial groups using diagnostic fatty acid indicators: bacteria, gram-negative bacteria, gram-positive, fungi, fungus, protozoon, methanogens I, methanogens, aerobic bacteria and anaerobic bacteria (Table S1).

We measured diameter at breast height (DBH, diameter at 1.3 m) using a tape measure (0.1-mm precision). We surveyed the plant species in each plot in August of 2017. We calculated species diversity according to *Leathwick, Burns & Clarkson (2010)*.

## Data processing and statistical analysis

We used SPSS 20.0 (IBM, Chicago, IL, USA) and R for statistical analyses. We considered each plot as an experimental unit, and the replicated data were averaged by plot for the analyses. Prior to conducting ANOVA analysis, all variables were checked for normal distribution (Kolmogorov–Smirnov test) and homogeneity (Levene's test). Then ANOVA analysis were analyzed in separate sampling season for the variables. Then ANOVA were performed to compare C and N pool stocks in the older and younger forests using Turkey's HSD test. We present all results as the mean value ± standard error. We considered $p < 0.05$ to be statistically significant. The data to examine relation between soil chemical variables were pooled from 10–20, 20–30, 30–40 and 40–50 cm depths and five sampling seasons and six independent plots (for each index $n = 150$) were examined Pearson relationships using R and the "Performance Analytics" package.

# RESULTS

## General condition of plant and soil of the sites

Trees were bigger in 1GF forests: diameter at breast height was 56.6% larger in 1GF than 2GF forests ($p = 0.002$), and average tree height was 11% higher in 1GF than the 2GF plots (Table 1). There were no significant differences in total phosphorus ($p = 0.08$), total potassium ($p = 0.84$), bulk density ($p = 0.44$) or mechanical composition of soil across the thinning treatments (Table 1) for samples collected in August 2016.

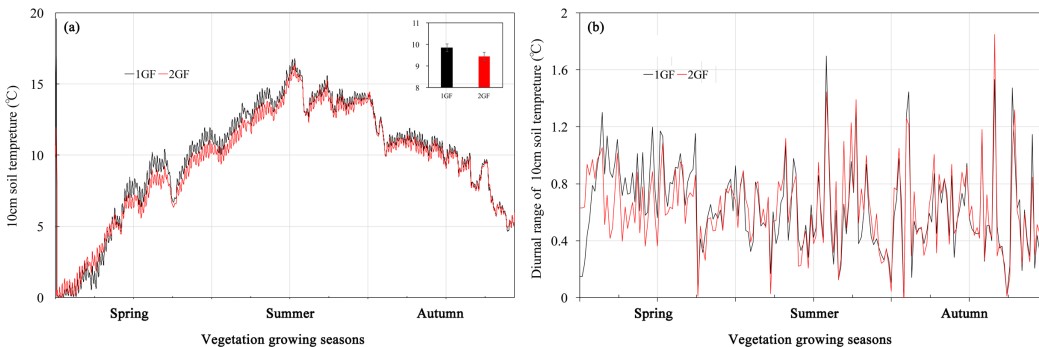

**Figure 1** Soil temperatures (A) and diurnal variation of temperature (B) at 10-cm depth in plots of 24-year-old (2GF) and 40-year-old (1GF) *Larix principis-rupprechtii* from spring to autumn in 2017. The gray bars on the time line are the sampling times.

Understory species composition was relatively similar in the two type forests. There were no significant differences in the diversity of shrub or herb layers ($p > 0.01$) as measured by species richness, species Shannon index, or species evenness (Table 2). The 1GF plots contained significantly thicker litter in April ($p = 0.041$) and October ($p = 0.009$) of 2017, and had 30% thicker litter across the four sampling seasons (Apr, Jun, Aug and Oct) of 2017.

## Soil temperatures

Soil temperature averaged across the vegetation growing season in the 1GF was 9.81 °C, significantly higher than in 2GF plots, 9.41 °C ($p < 0.001$). However, when the soil temperatures were below 4 °C, the 1GF plots were significantly cooler than 2GF, 0.13 °C compared to 0.22 °C. In May and June, temperatures in 1GF plots were 7.63% higher than in 2GF (Fig. 1). On July 24, 2017, the soil temperatures reached peak temperature in both forest types, 16.4 °C. In July, August, and September the soil temperatures were narrowly higher in 1GF (2.80% higher), and in October and later, soil temperatures in 1GF and 2GF were 7.10 and 7.06 °C (Fig. 1A). In summary, 10-cm soils were warmer in 1GF during the growing season, but cooler in winter. However, the diurnal variation of temperature was not influenced by forest age (Fig. 1B).

## Soil moisture and pH

Forests age significantly ($p < 0.001$) influenced pH and soil moisture (Table 3). 2GF forests had consistently higher pH values. Data pooled across the five sampling seasons indicated that pH value in the 2GF was 7% higher than 1GF and the soil moisture was 8% lower. In August and October 2017, soil moisture was significantly higher in 1GF than 2GF; 1GF soils contained 12.8% and 22.8% more moisture in August and October, respectively (Fig. 2B).

## PLFA

The composition of microbial community measured by PLFA was relatively consistent in the two forest types (Fig. 3): 74% of the microbiome composition was bacteria, 15% fungi, 10% antinomies, and 1% anhistozoa. There were no significant differences in

**Table 3 Results of the three-way ANOVA for the soil properties crossing plant growing seasons in 2017.** Tre: two Treatments; Sea: five sampling seasons; Dep: five soil depths, 10 cm for each soil layer.

| Fixed factors | Df | SOC F | SOC p | STN F | STN p | Moisture F | Moisture p | pH F | pH p | MBC F | MBC p | MBN F | MBN p | MBC/MBN F | MBC/MBN p | DOC F | DOC p | DON F | DON p | DOC/DON F | DOC/DON p |
|---|---|---|---|---|---|---|---|---|---|---|---|---|---|---|---|---|---|---|---|---|---|
| Tre | 1 | 1.217 | ns | 0.398 | ns | 20.854 | <0.01 | 43.338 | <0.01 | 20.517 | <0.01 | 25.176 | <0.01 | 11.902 | <0.01 | 12.65 | <0.01 | 25.258 | <0.01 | 43.718 | <0.01 |
| Sea | 4 | 2.597 | 0.041 | 1.013 | ns | 0.325 | ns | 0.462 | ns | 1.093 | ns | 26.278 | <0.01 | 1.852 | ns | 1.613 | ns | 0.117 | ns | 17.112 | <0.01 |
| Dep | 4 | 1.050 | ns | 9.295 | <0.01 | 0.087 | ns | 1.365 | ns | 0.204 | ns | 4.289 | <0.01 | 0.653 | ns | 1.623 | ns | 0.058 | ns | 2.563 | 0.043 |
| Sea × Tre | 4 | 6.668 | <0.01 | 0.626 | ns | 0.323 | ns | 1.406 | ns | 1.137 | ns | 3.155 | 0.017 | 1.852 | ns | 3.720 | <0.01 | 0.022 | ns | 4.202 | <0.01 |
| Dep × Tre | 4 | 3.359 | 0.013 | 0.086 | ns | 0.166 | ns | 0.163 | ns | 0.102 | ns | 0.882 | ns | 0.653 | ns | 0.200 | ns | 0.038 | ns | 0.362 | ns |
| Dep × Sea | 16 | 1.711 | 0.056 | 0.293 | ns | 0.013 | ns | 0.146 | ns | 0.042 | ns | 0.902 | ns | 0.192 | ns | 0.055 | ns | <0.01 | ns | 0.386 | ns |
| Dep × Sea × Tre | 16 | 1.778 | 0.045 | 0.047 | ns | 0.012 | ns | 0.225 | ns | 0.042 | ns | 0.452 | ns | 0.192 | ns | 0.040 | ns | <0.01 | ns | 0.523 | ns |

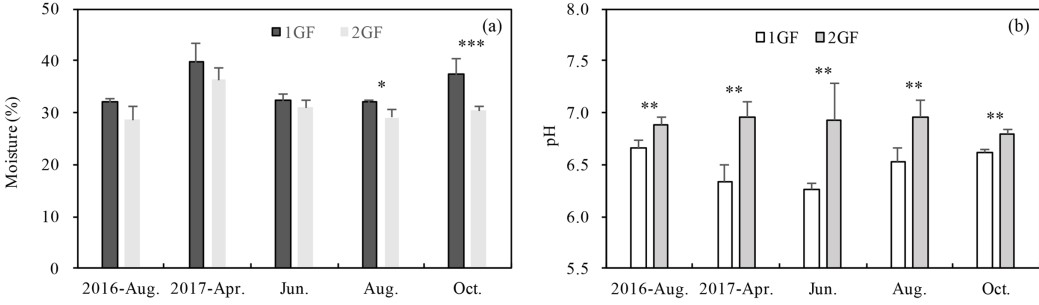

**Figure 2** Variation in soil moisture content (A) and soil pH (B) in 24-year-old (2GF) and 40-year-old (1GF) stands of *Larix principis-rupprechtii* forests across the plant growing seasons in 2017 and August 2016. Each value in the plot represents the average value of three plots replicates from five soil depths. The error bars represent the standard error and different letters indicate significant differences among treatments, $^*p < 0.1$; $^{**}p < 0.05$; $^{***}p < 0.01$.

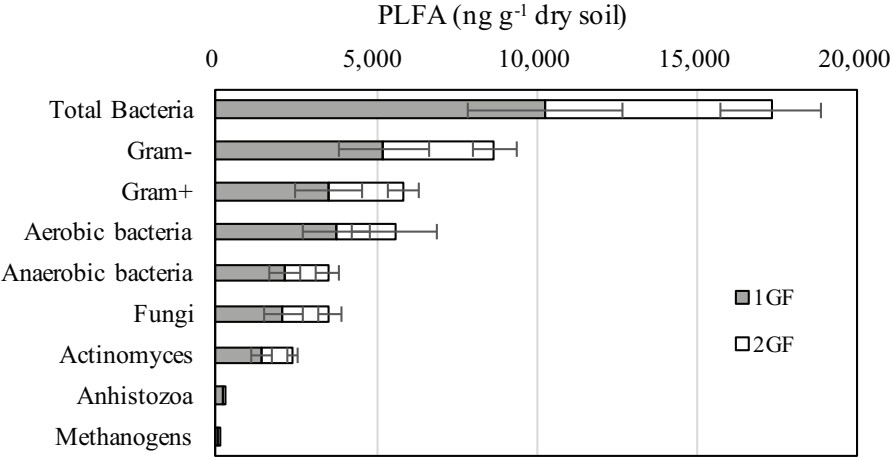

**Figure 3** Abundances of the dominant microbial communities for the 24 and 40 years old plantation soils from the 0–10 cm soils of August-2017. The relative abundances are based on the proportional frequencies of PLFA that could be classified. 1GF, the 24 years old *Larix principis-rupprechtii*forests; 2GF: 40 years old *Larix principis-rupprechtii* forests. Each value in the plot represents the average value of three plots replicates.

the composition of microorganisms in the two forests treatments when we analyzed Gram-positive or negative bacteria, methanogens, aerobic bacteria, or anaerobic bacteria. Detailed information about the microbial communities is available in Table S1.

## Soil total C and N pools

1GF forests had higher SOC and STN values on average, but the relationship differed across seasons. Three-way ANOVA indicated that SOC was affected by stand age, soil depths and seasons ($p < 0.05$). To test the factors separately, we measured C pools in different seasons independently. Soil organic pool (summed across five soil depths) in 1GF was 23% higher than in 2GF forests when the data were pooled across all the sampling seasons (1GF = 12.43± 1.73 kg m$^{-2}$; 2GF = 9.56 ± 1.03 kg m$^{-2}$). SOC values differed significantly ($p < 0.05$) between 1GF and 2GF forests in four of five sampling seasons; only June-2017 did not differ ($p > 0.01$, Fig. 4A).

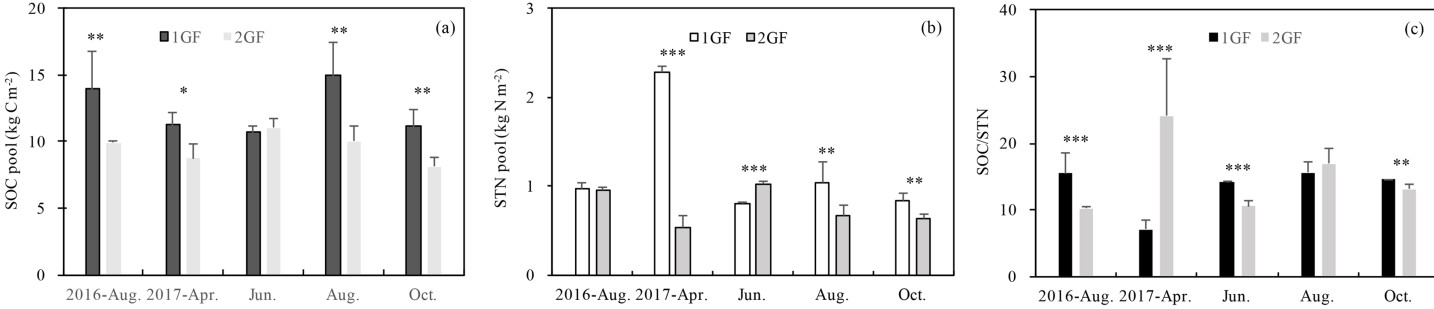

**Figure 4 Variation in soil organic C pool (A), soil total N pool (B), SOC/STN content (C) under the 24 years old *Larix principis-rupprechtii* forests and 40 years old forests across the plant growing seasons in 2017 and August 2016.** 1GF, the 24 years old *Larix principis-rupprechtii* forests; 2GF: 40 years old *Larix principis-rupprechtii* forests. Each value in the plot represents the average value of three plots replicates. The error bars represent the standard error and different letters indicate significant differences among treatments, $^*p < 0.1$; $^{**}p < 0.05$; $^{***}p < 0.01$.

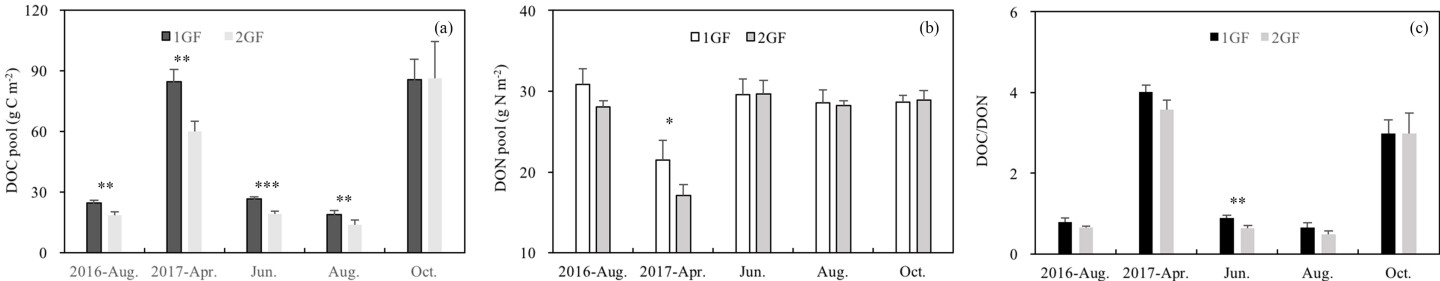

**Figure 5 Variation in soil dissolved organic C pool (A), soil dissolved N pool (B), DOC/DON (C) under the two different stand age forests across the plant growing seasons in 2017 and August 2016.** 1GF, the 24 years old *Larix principis-rupprechtii* forests; 2GF: 40 years old *Larix principis-rupprechtii* forests. Each value in the plot represents the average value of three plots replicates. The error bars represent the standard error and different letters indicate significant differences among treatments, $^*p < 0.1$; $^{**}p < 0.05$; $^{***}p < 0.01$.

The STN pool was also significantly affected by stand age ($p < 0.001$, Table 3). STN pool in 1GF was 16% higher when data were pooled across all sampling seasons (1GF = 2.07 ± 0.97 kg m$^{-2}$; 2GF = 1.22 ± 0.30 kg m$^{-2}$). In June 2017, the STN pool was significantly higher in 2GF: 1GF (1.30 ± 0.03) < 2GF (1.56 ± 0.13). In August 2016 and October 2017, the differences were not statistically significant (Fig. 4B).

The ratio of soil organic C to soil total nitrogen (C/N) differed significantly by age treatments in August 2016 ($p = 0.081$), June 2017 ($p = 0.048$), and October 2017 ($p = 0.032$); the ratio was lower in 2GF forests (Fig. 4C).

## Soil active C and N components

The DOC content was 18% higher in 1GF when the data were pooled across all sampling seasons (1GF = 47.97 g C m$^{-2}$; 2GF = 39.56 g C m$^{-2}$), and DOC values showed significant interaction between stand age and seasons ($p < 0.001$). Analyzing each season separately, the DOC pool was affected by the stand age in four seasons out of five (Fig. 5A). The DOC pools were four times larger in in April and October, the start and end of the growing season, than during the growing season (June and August) ($p < 0.001$) (Fig. 5A).

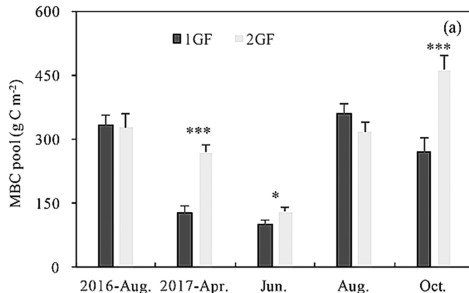
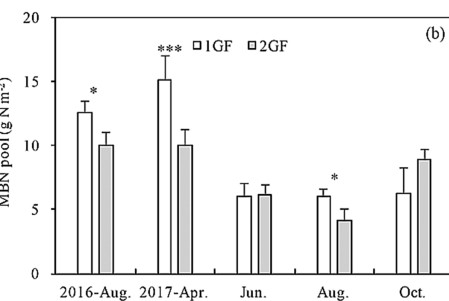
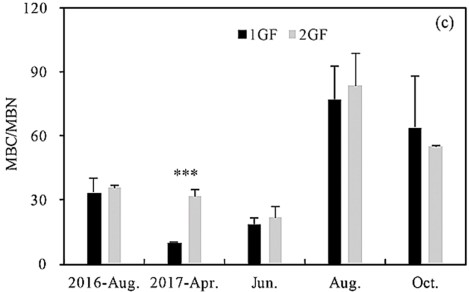

**Figure 6 Variation in soil microbial biomass C pool (A), soil microbial biomass N pool (B) and MBC/MBN content (C) under the two different stand age forests across the plant growing seasons in 2017 and August 2016.** 1GF, the 24 years old *Larix principis-rupprechtii* forests; 2GF: 40 years old *Larix principis-rupprechtii* forests. Each value in the plot represents the average value of three plots replicates. The error bars represent the standard error and different letters indicate significant differences among treatments, $*p < 0.1$; $***p < 0.01$.

1GF and 2GF forests also differed significantly in DON content ($p < 0.001$, Table 3). DON pool was 5% higher in 1GF when the data were pooled across sampling seasons (1GF = 27.83 g N m$^{-2}$, 2GF = 26.38 g N m$^{-2}$). In April 2017 DON was lower than in other seasons (Fig. 5B). DOC/DON ratio differed between the two age treatments only in June 2017 ($p = 0.021$, Fig. 5C), with complicated interaction effects from stand age, seasons, and soil depths (Table 3).

MBC content was 14% lower in 1GF when the data were pooled across sampling seasons (1GF = 9.21 g C m$^{-2}$; 2GF = 7.88 g C m$^{-2}$). MBC content differed significantly between the two forest age treatments ($p < 0.001$) (Table 3). When the different sampling seasons were analyzed separately, the MBC pool differed significantly between age treatments in April 2017 ($p < 0.001$), June 2017 ($p = 0.049$), and October 2017 ($p = 0.005$). We found lower MBC pools early in the growing season—April and June 2017 (Fig. 6A).

MBN content in 1GF was 24% lower than 2GF when the data were pooled across sampling seasons (1GF = 7.39 mg N m$^{-2}$; 2GF = 5.61 mg N m$^{-2}$) (Fig. 6B). MBN values also varied across soil depths and seasons ($p < 0.001$) (Table 3). MBN values peaked in April 2017. The MBN/MBC ratio was significantly higher in 2GF forests when the soil temperatures were warming from April to August (Fig. 6C).

## Soil respiration

After incubating the soil samples collected in August 2017, cumulative C mineralization did not differ between the two forest age treatments ($p > 0.05$, Fig. 7A). Cumulative C mineralization ranged from 2.28 ± 0.14 (in 1GF) to 2.19 ± 0.06 g C kg$^{-1}$ soil (in 2GF, Fig. 7A).

Mineralized C (MC), expressed on a per kg total soil C basis (rather than per kg soil), which is a means to normalize against differences in soil type (total C) between the two forest age treatments, was significantly greater at 2GF than at 1GF ($p < 0.001$, Fig. 7B). MC differed from 112.6 ± 9.5 to 154.5 ± 23.6 mg C g$^{-1}$ soil C in 1GF sites compared to 154.5 ± 23.6 mg C g$^{-1}$ soil C at the 2GF site in August 2017 after incubating (Fig. 7B).

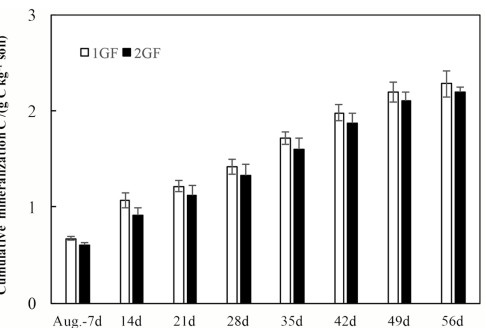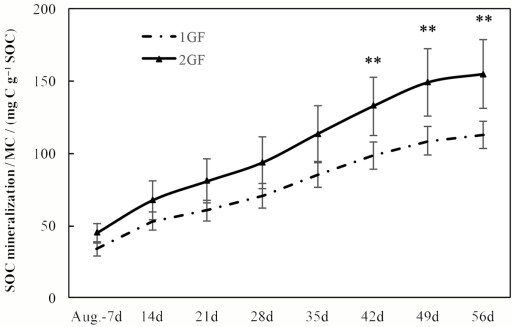

**Figure 7 Cumulative CO$_2$ emissions from soils (0–50 cm depth) over time (A) and the SOC mineralization rates (B) from croplands and poplar plantation soil samples during the 56-day incubation study.** Systems with ** are statistically different at $p < 0.05$ ($n = 3$).

## DISCUSSION

We explored potential mechanisms driving differences between 24-year-old (2GF) and 40-year-old (1GF) forest stands dominated by *Larix principis-rupprechtii*. Overall we found that older forests accumulated 24% more soil C than did younger forests. Below, we discuss the factors associated with this difference in C storage.

### Environmental factors

Soil temperatures were higher in older 1GF forests relative to 2GF forests during the plant growing season. The direction of causality between temperature and stand structure is not clear; significantly higher DBH, crown breadth, and lower under-branch height in 1GF forests may have contributed to higher soil temperatures, or they may have resulted from higher soil temperatures. Soil warming studies have shown direct influence of soil temperature on stem growth (*Lupi et al., 2012*). Thicker litter cover and microbe activity in older 1GF forests may also have contributed to higher soil temperatures and moisture, as thicker litter could insulate soils and prevent temperature and moisture from dissipating through the atmosphere, and microbial decomposition could generate more heat. Higher temperatures could then increase the activity of decomposers, generating more heat and indirectly affect decomposition through other ecosystem feedbacks (*Allison & Treseder, 2011*).

Older 1GF forests also contained more soil moisture than did younger 2GF forests. Soil moisture is important for the growth of plants roots (*Gaines et al., 2016*), underground root respiration, and decomposition of litter by microorganisms (*Liu et al., 2006*) which are the important sources C. Microbial communities did not appear to differ significantly between the forest age treatments, so the amount of microbes did not likely play an important role in the differences in C storage between treatments.

Soil pH was significantly higher in the younger 2GF forests and was negatively related to C and N components (Fig. 8). A study conducted in North America indicated that 50–70-year-old forest soils had higher concentrations of Mg, Ca, NO$_3^{-1}$ and a higher pH than

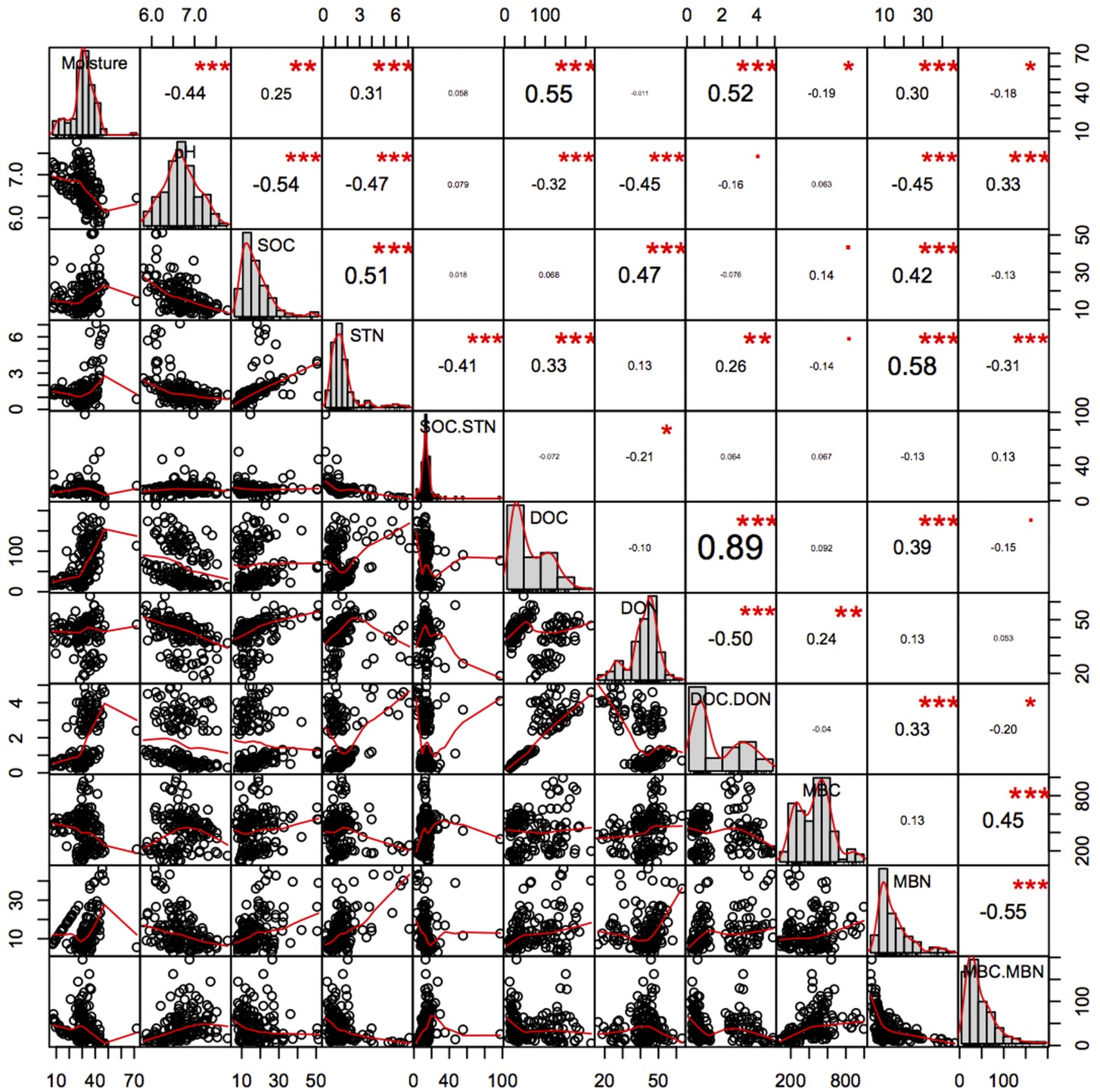

**Figure 8 Pearson relationship of different soil properties across forests age, seasons and soil depths.** $n = 150$, i.e. two stands age treatments *five seasons *three repeats *five soil depths. $^*p < 0.1$; $^{**}p < 0.05$; $^{***}p < 0.01$.

120–150-year-old forest soils, which may have been an age effect (*Yesilonis et al., 2016*). The combination of years of soil leaching and lack of leaf litter input high in nutrients such as calcium could partly explain the lower pH in the 1GF (*Yesilonis et al., 2016*). Recent

research has indicated that lower pH could promote C storage and plant growth (*Chen et al., 2018*).

## Soil C and N

The SOC pool is the balance between the C input from aboveground litterfall and belowground rhizo-deposition, and release by decomposition as $CO_2$ (*Jandl et al., 2007*). In this study older 1GF forests had higher SOC pools and relatively higher STN pools, indicating that older 1GF forests held more C across seasons. Increasing STN could impact the sustainability of C sinks in forest ecosystems (*Townsend et al., 1996*), as a result of interactions between the C and N cycles (*Rastetter, Ågren & Shaver, 1997*). Studies have suggested that STN is frequently limiting and that more N could contribute to C accumulation and greater microbe activity. Here, we found a significant positive correlation between SOC and STN (Fig. 8), suggesting that N may contribute to C accumulation. Soil pH was also significantly negatively associated with SOC. The thicker litter accumulation was an important source for soil organic matter in 1GF plots.

## Soil C processes

Active C pools are widely studied in different forest management areas because they can reflect C stock dynamics (*Franzluebbers & Arshad, 1997*). Soil active organic C pools (DOC, MBC, MC) provide effective sources of C for soil microorganisms and are easily decomposed into $CO_2$ (*Yang et al., 2009*; *Liang et al., 2012*). Generally, we measured higher active C components in the older 1GF plots, which also had larger soil C pools than did younger 2GF forests.

Soil C pools are mainly influenced by the balance between input from plants and loss from leaching of active C components like DOC (*Liying et al., 2015*) or the release of $CO_2$ (*Creamer, Filley & Boutton, 2001*). It is generally believed that DOC increases with seasonal increases in soil temperature and enhanced microbial activity (*Bonnett, Ostle & Freeman, 2006*), while here, we found a decrease in the DOC while temperatures rose across the plant growing season. MBC was negatively related to DOC; MBC increased when DOC decreased. DOC content may be derived from litter leachates, root exudates, or microbial degradation products (*Ohno, Griffin & Liebman, 2005*) and feed soil microorganisms. In many studies, DOC is positively correlated with SOC, which is similar to what we found here—soil in the older 1GF forests contained higher DOC and SOC values. Generally, pH was negatively related to the N pools, DON and MBN and STN, and moisture was positively related to them (Fig. 8); however, the active C components were not significantly related to pH. Generally, more liable C may have accumulated in older 1GF forests because of more litter, and increases in active C pools could result in accumulation in recalcitrant C pools, which could contribute to greater C storage in 1GF forests.

Cumulative C mineralization expressed on a per kg total soil C basis (MC, Cumulative C mineralization/SOC) was significantly higher in 1GF plots after incubating for 56 days. One potential explanation for the higher C stock in the older 1GF forests is that soils in younger 2GF forests release more C through $CO_2$. Organic C mineralization reflects the

transformation of organic matter to inorganic chemical matter, deposed by microorganisms, and producing $CO_2$ or carbonate. $CO_2$, a gas, is more likely to leave the soil and cause the C stocks to fall. Increased MC could also result from greater microbial activity in 2GF forests in April and October 2017.

Increases in the active C pool could eventually result in greater accumulation of C in more recalcitrant pools, as the increased active C pool will transfer into the more recalcitrant pools of C through physical breakdown of organic material and microbial mediated processes (*Grandy & Neff, 2008*; *Sprunger & Robertson, 2018*). Here more liable C was measured in the older 1GF forests; greater litter cover in the 1GF forests increased the nutrient input and could have been the source for the liable C pools. The microbe activity in the younger 2GF forests was likely responsible for the significantly higher MC in the 2GF, as we measured here. Higher MBC reflected more microorganisms in 2GF forests. The higher pH in 2GF forests may have been caused by the other soil mineralization product, carbonate. Younger 2GF forests had higher microbial activity and MC, possibly caused by other soil organic material mineralization products.

Modeling of forests in eastern United States indicates that including anthropogenic disturbance improves model accuracy in simulating C stocks and fluxes of eastern temperate forests (*Dangal, Felzer & Hurteau, 2014*). The study also suggests that after an initial release of carbon following forest harvest, forest ecosystems are able to retain or even gain carbon in plants and soils following planting. Future work will focus on how environmental factors, active C pools and soil carbon pools change across more diverse forest ages.

## CONCLUSION

Our results demonstrate that for the purpose of increasing C storage, 40-year-old *Larix principis-rupprechtii* forests function better than 24-year-old forests. The reduction in C storage in the younger forests results from environmental factors, active C pools, and microbe activity. The older 1GF forests contain greater C stocks and active C components due to reduced C release through mineralization, and less loss of C as $CO_2$. The C mineralization expressed on a per kg total soil C in the 24-year-old forests produced more $CO_2$ after incubating soils for about 56 days, especially during the plant growing season. Microbe activity appears to have driven C loss in the younger 2GF forests. MBC (C/N and PLFA) indicate that the amount and activity of microorganisms are greater in the 2GF forests. And MBC was closely related to MC—higher MBC was associated with higher MC.

We recommend that maintaining 40-year-old *Larix principis-rupprechtii* forests is a better strategy for storing C than maintaining 24-year old larch plantations. We plan to study further whether C stocks might decline as forests continue to age.

## ACKNOWLEDGEMENTS

We gratefully acknowledge the support from the Taiyue Forestry Bureau and the Haodifang Forestry Centre for fieldwork. We would like to thank Dr. Abe Miller-Rushing for his assistance with English language and grammatical editing of the manuscript.

### Funding

The study was supported by the National Key Research and Development Program of China (2016YFD0600205). The funders had no role in study design, data collection and analysis, decision to publish, or preparation of the manuscript.

### Grant Disclosures

The following grant information was disclosed by the authors:
National Key Research and Development Program of China: 2016YFD0600205.

### Competing Interests

The authors declare that they have no competing interests.

### Author Contributions

- Junyong Ma conceived and designed the experiments, performed the experiments, prepared figures and/or tables, authored or reviewed drafts of the paper, and approved the final draft.
- Hairong Han conceived and designed the experiments, analyzed the data, prepared figures and/or tables, and approved the final draft.
- Xiaoqin Cheng analyzed the data, authored or reviewed drafts of the paper, and approved the final draft.

### Data Availability

The raw data is available as a Supplemental File.

### Supplemental Information

Supplemental information for this article can be found online at http://dx.doi.org/10.7717/peerj.8384#supplemental-information.

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
