# Peer review of "Soil temperatures and active carbon components as key drivers of C stock dynamics between two different stand ages of Larix principis-rupprechtii plantation"

_PeerJ, doi:10.7717/peerj.8384_

## Round 0.1 · original submission · Major Revisions

It seemed to the reviewers that, despite potential interests of this study, the manuscript is still unclear and not mature enough yet. Please revise according to the comments from the two reviewers.

[]

Reviewer 1 ·

Basic reporting

The authors examine the impacts of stand age on soil carbon in a Larch forest. The authors seek to assess differences in C stocks in stands with trees that are 40- and 24-year-old. They also examine variability in soil and microclimate conditions and potential explanations for differences between stands. The description and presentation of the results should be more specific to address all four study objectives outlined by the authors. Further work is necessary to ensure that conclusions are direct and clear.

General comments:
1. There are some areas where some clarification is needed to better convey scientific terminology throughout all sections. For example, in lines 69-70 “environmental variables C pool is affected by the dynamic of actively C components”, it is unclear what the components is being referred to and how they are varying temporally. There are several other areas that use similarly vague terminology. I’ve included some other lines in the specific comments.
2. While the authors set out with 4 study objectives, the importance of these objectives and the background can be hampered by too superficial of sentence phrasing. Throughout the introduction, there are often sentences that often indicate the importance of a process without providing context. For example, line 45 indicates “Many forests, especially plantation forests, are managed for particular age structures.” This does not help me understand whether I should expect the younger or older stand in the study to reflect management practices. It is hard for the reader to gather the importance of the study. Other examples include lines 45-47, 59-60, 67-68, 77-78.
3. The discussion would benefit from a clearer interpretation of the results that relates back to the big picture and original objectives. More citations would be useful. For example, lines 270-282 have a series of competing explanations for the impacts of soil and microclimatic conditions of soil carbon. It can be a little confusing for remembering the differences between the stands and the main take away.
4. The study would benefit from clear, direct conclusions (lines 333-44). The conclusions section largely repeats results and provides open ended, vague explanations for carbon dynamics in the two stands. It is difficult to keep track of what the “difference in crown cover” or “difference in litter input” is for the two stands (lines 343-344). Furthermore, objective 4 (evaluate strategies for storing C in managed forests) is not discussed entirely and there are no conclusions about strategies. This objective seems particularly useful for providing more context for the study.

Specific comments: Please clarify phrasing with more detail and clear definition.
Line 57: “combination of factors”. What factors?
Line 59: “interactions between human activities and environmental factors and their effects on forest function and dynamics” What interactions and effects?
Line 292: “different seasons” What seasons?
Line 335: clearly state how and what environmental factors, active C pools impacted carbon

Experimental design

Overall the experimental methods and objectives are sound. The authors’ data and analysis would be useful for understanding the carbon balance in managed forest plantations. However, some issues with basic reporting hamper the reporting of the research and its relevance.

Validity of the findings

The results and conclusions could be clarified and more specific. The data provided is well organized, but needs a little more clarification in the metadata.

General comments
1. The full statistical test should be reported not just the p value. For example, it would be good to see the test statistic (ex, t-statistic with degrees of freedom) in addition to the p value for open and thorough reporting of statistical analysis. It would also be good to use a consistent number of significant digits. In some areas, p values are reported as <0.01 and others p=0.009. It would also be helpful include the mean in some comparisons rather than percent differences (for example lines 192-200, line 212).

2. The provided data has a column that identifies each sample does not have any explanation for each identifier. It appears there is an indication of stand age, stand replicate, and sample replicate. However, it is not possible to confirm this and stand/age replicates may be confounded depending on user interpretation. I recommend breaking identifiers up into separate columns and having a brief explanation for what each identifier represents.

3.There is also no PLFA mass data included.

Specific comments:
Table 1: error in numbers. Trees are listed as >2000 meters in height.
Line 270-2782: Need to expand on reasoning behind the crown and branch height influences on soil temperature. There are also many explanations for the differences in soil temperature. While the temperature is significantly different, there is little functional difference in the difference between the mean soil temperature (0.4 C), less than half of the typical diurnal range in temperature for in both stands. The difference appears to be driven by the spring, but the seasonality and implications for soil carbon are not discussed. The higher soil moisture discussed in the 278-282 would increase thermal conductivity, but lower thermal conductivity as a result of litter is a line of reasoning for the higher soil temperature.
In figure 8: correlations are shown in cases where high correlation is expected. For example, DOC is used directly in the calculation of ratio of DOC/DON and thus there is a high correlation. Other significant correlations often have a huge variability and vary non-linearly. Units would also be helpful.

Reviewer 2 ·

Basic reporting

Ma, and colleagues investigate changes in several soil properties including soil organic carbon in forest ecosystem across two different stand ages. In particular, the study compared 40-year old growth forest against 24-year-old plantation of Larix species in North China to show that old growth forest tend to store more carbon, which is likely associated with low microbial activities in old growth forests. In addition, the study found that soil temperature and soil moisture content were higher in old growth forests. While quantifying the carbon content of soil across different forest age is an important topic for understanding ecosystem health and climate change mitigation, there are several study limitation that need to be addressed prior to publication. In addition, the English language need to be thoroughly revised since I found several typos and grammatical errors.

Major comments:
1. While the study report differences in several soil attributes (organic carbon, pH, soil temperature, microbes etc) between two forest types with the findings that old growth forest are more resilient and store more carbon compared to planted forests, the mechanisms driving these changes are not well justified and explained. It seems abit counterintuitive to think that soil moisture, soil temperature and soil nitrogen would increase in old growth forest but still be able to hold for carbon in soils. One of the argument as suggested by the authors was that more C were accumulated in recalcitrant pools which are less vulnerable to microbial activity. While I agree with the authors justification, it is unclear how they determine the amount of C stored in recalcitrant pool. Since old growth forest have high litter input compared to planted forests, which could lead to more organic matter retention in labile pools leading to higher rate of decomposition.
2. It is unclear why there would be such a huge difference in the pH of the soil between the old growth and planted forest (see figure 2b), particularly in the month of June. Since both soil contain same species and given that the environmental conditions are quite similar, there is no justification of these difference. It is also unclear how the increase/decrease in pH would play an important role in ecosystem management (i.e. through increase or decrease in microbial activity).
3. The authors also simply report findings from their experimental studies and never try to put the outcomes of the study in the larger context. I think the authors should flesh out how/whether retaining the old growth forests is better for climate change mitigation compared to regrowing forests. For example, there are several studies that suggest that after an initial release of carbon following forest harvest, forest ecosystems are able to retain or even gain more carbon in plant and soil following plantation (see Dangal et al. 2014). However, the authors fail to include the legacy effects of forest harvest and regrowth on net carbon sequestration in soils.
4. The results of this study show that both temperature and soil moisture increase in old growth forests. However, the authors fail to clearly provide a justification of why both temperature and soil moisture increase. I believe that if there is an increase in temperature, there should be less soil moisture as an increase in temperature increase the vapor pressure deficit increasing the atmospheric demand of water vapor.
5. The writing need to be thoroughly revised. There are several grammatical errors that can be improved by contacting a native speaker to comment/revise the manuscript. Also, I found that there are Chinese language inserted in figure captions (please see figure 2).


Other comments
Ln 27: Is it a good idea to report temperature increase in percent? It is really surprising to me that a 2.9% increase in temperature was significant?
Ln 35: 3300 Pg C is not an accurate number. There is actually a large uncertainty in the total carbon stored in soil that could range from 411-2111 PgC (see Tian et al. 2015).
Ln 41: I disagree if forest soil store the largest amount of carbon. How about wetlands or even grasslands?. It is really important to know if you are talking about per unit area storage or total storage.
Ln 49-52: I do not think this study is focused on understanding the effects of different management in forest carbon. Is this paragraph even important?
Ln 69. Please revise this sentence. For example: “Forest C pools can also be affected by the dynamics of active C components”.
Ln 69: revise actively to active.
Ln 212-214: There is only a result of soil pH changes but no mention of soil moisture changes.
Ln 334: correct “retults” to “results”


Dangal et al. (2014) Effects of agriculture and timber harvest on carbon sequestration in the eastern US Forest. JGR-Biogeosciences
Tian et al. (2015) Global patterns and controls of soil organic carbon dynamics as simulated by multiple terrestrial biosphere models: Current status and future directions

Experimental design

Since the authors are comparing planted vs old growth forest, the experimental design does not apply

Validity of the findings

This is a reporting paper with a strong measurement data. So, the findings are valid but the authors need to put those findings in the larger context by comparing with similar results from other studies

Additional comments

see basic reporting

---

## Round 0.2 · accepted · Accept

I think you addressed all the comments from the two reviewers and your datatset it definitely worth publication.